# Global, regional, and national survey on burden and Quality of Care Index (QCI) of orofacial clefts: Global burden of disease systematic analysis 1990–2019

Ahmad Sofi-Mahmudi[1,2,3]*, Erfan Shamsoddin[4], Sahar Khademioore[2], Yeganeh Khazaei[5], Amin Vahdati[6], Marcos Roberto Tovani-Palone[7]

1 National Pain Centre, Department of Anesthesia, McMaster University, Hamilton, ON, Canada, 2 Department of Health Research Methods, Evidence and Impact, McMaster University, Hamilton, ON, Canada, 3 Seqiz Health Network, Kurdistan University of Medical Sciences, Seqiz, Kurdistan, 4 Cochrane Iran Associate Centre, National Institute for Medical Research Development (NIMAD), Tehran, Iran, 5 Department of Statistics, Statistical Consultation Unit, StaBLab, LMU Munich, Munich, Germany, 6 Centre for Biostatistics, University of Manchester, Manchester, United Kingdom, 7 Ribeirão Preto Medical School, University of São Paulo, São Paulo, Brazil

* sofima@mcmaster.ca, a.sofimahmudi@gmail.com

**Data Availability Statement:** All code and data related to the study were shared via its OSF

## Abstract

### Background

Orofacial clefts are the most common craniofacial anomalies that include a variety of conditions affecting the lips and oral cavity. They remain a significant global public health challenge. Despite this, the quality of care for orofacial clefts has not been investigated at global and country levels.

### Objective

We aimed to measure the quality-of-care index (QCI) for orofacial clefts worldwide.

### Methods

We used the 2019 Global Burden of Disease data to create a multifactorial index (QCI) to assess orofacial clefts globally and nationally. By utilizing data on incidence, prevalence, years of life lost, and years lived with disability, we defined four ratios to indirectly reflect the quality of healthcare. Subsequently, we conducted a principal component analysis to identify the most critical variables that could account for the observed variability. The outcome of this analysis was defined as the QCI for orofacial clefts. Following this, we tracked the QCI trends among males and females worldwide across various regions and countries, considering factors such as the socio-demographic index and World Bank classifications.

### Results

Globally, the QCI for orofacial clefts exhibited a consistent upward trend from 1990 to 2019 (66.4 to 90.2) overall and for females (82.9 to 94.3) and males (72.8 to 93.6). In the year

repository (https://osf.io/r94ch/) and GitHub (https://github.com/choxos/qci-cleft) at the time of submission of the manuscript.

**Funding:** The author(s) received no specific funding for this work.

**Competing interests:** The authors have declared that no competing interests exist.

2019, the top five countries with the highest QCI scores were as follows: Norway (QCI = 99.9), Ireland (99.4), France (99.4), Germany (99.3), the Netherlands (99.3), and Malta (99.3). Conversely, the five countries with the lowest QCI scores on a global scale in 2019 were Somalia (59.1), Niger (67.6), Burkina Faso (72.6), Ethiopia (73.0), and Mali (74.4). Gender difference showed a converging trend from 1990 to 2019 (optimal gender disparity ratio (GDR): 123 vs. 163 countries), and the GDR showed a move toward optimization (between 0.95 and 1.05) in the better and worse parts of the world.

## Conclusion

Despite the positive results regarding the QCI for orofacial clefts worldwide, some countries showed a slight negative trend.

## Introduction

Orofacial clefts are the most common craniofacial anomalies that encompass a variety of conditions affecting the lips and oral cavity. They are estimated to affect 1 in every 700 births, posing a significant global public health challenge [1–4]. Clefts can be unilateral or bilateral, complete or incomplete, affecting the lip, palate, or both [5]. The underlying causes of these conditions largely remain unidentified [6] since the etiology of clefts is complex and a variety of genes with variable functions were shown to be involved in clefts' occurrence.

Effects on speech, hearing, appearance, and cognition can lead to long-lasting adverse outcomes for health and social integration [6]. Affected children need multidisciplinary care from birth until adulthood and have higher morbidity and mortality throughout life than unaffected individuals [7, 8]. Also, many children and their families are affected psychologically to some extent [9]. Care for children born with these defects generally includes a multitude of disciplines, including nursing, plastic surgery, maxillofacial surgery, otolaryngology, speech therapy, audiology, counselling, psychology, genetics, orthodontics, and operative dentistry. Due to this complexity, in both developing and developed countries, there are ongoing concerns regarding the standards of care for patients with cleft lip, cleft lip and palate, or cleft palate alone [10, 11].

The Global Burden of Disease (GBD) 2019 study estimates that the global disability-adjusted life years (DALYs) rate for orofacial clefts has decreased by more than 55% since 1990 [12]. In order to assess and quantify this progress achieved in addressing the global burden of orofacial clefts, the introduction of a comprehensive metric like the Quality-of-Care Index (QCI) could play a pivotal role [13]. The QCI offers a structured framework to evaluate the effectiveness and adequacy of healthcare interventions, considering factors such as access to medical services, surgical outcomes, patient satisfaction, and follow-up care. By incorporating the QCI into the evaluation framework, policymakers and healthcare professionals can gain valuable insights into the quality of care provided to individuals with orofacial clefts over time.

In this study, we aim to estimate the quality of care for orofacial clefts between 1990 and 2019. This new approach could address the existing gap in data availability and enable a more nuanced understanding of the disparities in orofacial cleft care across different countries and regions, and contribute to the enhancement of evidence-based policymaking for orofacial cleft management.

## Methods

The protocol of this descriptive study was published beforehand on the Open Science Framework (OSF) website (https://osf.io/94wr7). All code and data related to the study were shared via its OSF repository (https://osf.io/r94ch/) and GitHub (https://github.com/choxos/qci-cleft) at the time of submission of the manuscript. To ensure transparency and facilitate the reproducibility of our analyses, a PDF document containing the codes and corresponding outputs is provided in S1 Appendix.

The data used in this research was obtained from the GBD 2019 study, making it accessible to the public via the website http://ghdx.healthdata.org/gbd-2019. Identification of orofacial clefts in this analysis adhered to the International Statistical Classification of Diseases and Related Health Problems, 10th revision (ICD-10) as Q37. The GBD 2019 study offers a uniform method to calculate different health measures. These epidemiological measures encompass incidence, prevalence, premature mortality rates, years lived with disability (YLDs), and disability-adjusted life years (DALYs). The latter metric, DALYs, provides a composite indicator of population health burden, integrating both mortality and non-fatal health outcomes. These metrics are sorted by cause, age, gender, year, and where people live. Detailed information about the GBD study, including its inputs, analytical methods, results, and specific approaches for different causes, can be found in other sources [12].

Four indices related to the quality of care are described as follows:

1. Prevalence to incidence ratio $= \frac{\text{Prevalence}}{\text{Incidence}}$. This ratio illustrates the connection between the existing cases of the condition (prevalence) and the occurrence of new cases (incidence). A lower ratio could suggest either more efficient care or, conversely, a potential reduction in the lifespan of individuals with orofacial clefts.

2. Mortality to incidence ratio $= \frac{\text{Death}}{\text{Incidence}}$. This ratio serves as an indicator of the effectiveness of care provision, and a lower value indicates better care outcomes.

3. DALYs to prevalence ratio $= \frac{\text{DALYs}}{\text{Prevalence}}$. This ratio provides an assessment of the disease's overall burden, considering both mortality and morbidity. A higher ratio signifies a more substantial burden on the affected population.

4. YLLs to YLDs ratio $= \frac{\text{YLLs}}{\text{YLDs}}$. This ratio reveals the mortality impact of the disease, and higher values indicate a compromised survival status for individuals with orofacial clefts.

Due to challenges in accurately determining the number of orofacial malformations in pregnancies that end in spontaneous abortion, it is impossible to measure the true incidence of cleft palate alone and cleft lip with or without cleft palate. Consequently, birth prevalence is the preferred metric for assessing the frequency of orofacial clefts at the time of birth [14]. As a result, we incorporated prevalence into our index to achieve precise estimations for the Quality of Care Index (QCI) across various countries and regions.

To standardize these indices, we employed principal component analysis (PCA) as a multivariable analytical method. This approach derives linear combinations of variables as orthogonal or uncorrelated components [15]. Within this framework, the PCA's top-ranked component, formed as a linear combination of all variables, encompasses most of the information from these variables and is identified as the QCI. QCI scores fall within a range of 0 to 100, with higher scores indicating a superior status. PCA plays a pivotal role in simplifying the intricate relationships among these indices, thus aiding in a comprehensive evaluation of their combined influence on the QCI. The QCI has previously been implemented in several other publications, including lethal [13, 16–18] and non-lethal [19–21] conditions.

To assess the distribution, we utilized the Socio-demographic Index (SDI), which is a concise measure reflecting the developmental status of a region. This evaluation takes into account the rankings of average per capita incomes (Purchasing Power Parity or PPP), levels of educational attainment, and fertility rates across all areas included in the GBD study. Additionally, we integrated relevant World Bank classifications for global regions and/or countries. We used quintiles in a given year to categorize countries based on their QCI scores.

To assess gender inequality within each country, we utilized the gender disparity ratio (GDR), determined by comparing the QCI between males and females. We divided these ratios into five groups or quintiles, categorized as follows: 0 to 0.5, 0.5 to 0.95, 0.95 to 1.05, 1.05 to 1.5, and above 1.5. The quintile ranging from 0.95 to 1.05 was identified as the optimal GDR category. Full details of the analytical methods used in this study can be found elsewhere [22].

All the maps were created using the *rnaturalearth* package [23] in R [24], which uses the Natural Earth, (https://www.naturalearthdata.com/), a public domain map dataset including vector country and other administrative boundaries.

Starting now, all the DALY rates and QCIs mentioned in this paper, except for the absolute values, are presented as age-standardized figures.

### Validation

To validate our findings, we employed a mixed-effect regression model where the dependent variable was the QCI. The independent variables encompassed healthcare utilization for both inpatient and outpatient services, orofacial cleft-related mortality, prevalence of the condition, and deaths attributed to all risk factors [17].

In our analysis, countries were treated as random effects, and we determined a Pearson's correlation coefficient of 0.6 between the predicted QCI and the Healthcare Access and Quality of Care Index (HAQI), which assesses healthcare service accessibility [18]. Approximately 82% of countries had available utilization data, with lower availability in low-income countries (around 65% coverage). All statistical analyses were performed using R 4.2.2. A comprehensive explanation of our mathematical model's procedures and statistical methodology can be found in another source [19].

## Results

### Burden

On a global scale, the age-standardized rate of DALYs attributed to orofacial clefts was 19.63 per 100,000 (with a 95% uncertainty interval (UI) ranging from 12.85 to 27.44) in the year 1990. This rate was higher among males, standing at 21.04 per 100,000 (95% UI: 10.45–35.28), compared to 18.14 per 100,000 (95% UI: 12.12–35.68) among females during the same year.

Over the course of time, these numbers have consistently decreased, resulting in an average overall DALYs rate of 7.51 per 100,000 (95% UI: 5.10–11.57) in the year 2019. Specifically, in 2019, the DALYs rate was 7.58 per 100,000 (95% UI: 5.03–13.44) for males and 7.45 per 100,000 (95% UI: 4.84–12.73) for females.

The highest reduction in DALYs rate was observed in the upper-middle-income countries according to the World Bank classification, with a decrease of 83.2%, as well as in high-middle SDI countries, with a reduction of 82.6%. All regions across the world have experienced a decline in DALYs rate since 1990.

### Quality of care index and gender inequity

The global QCI for orofacial clefts showed a consistent increase from 1990 to 2019, climbing from 66.3 to 90.2. In 1990, the QCI was higher for females compared to males, with scores of

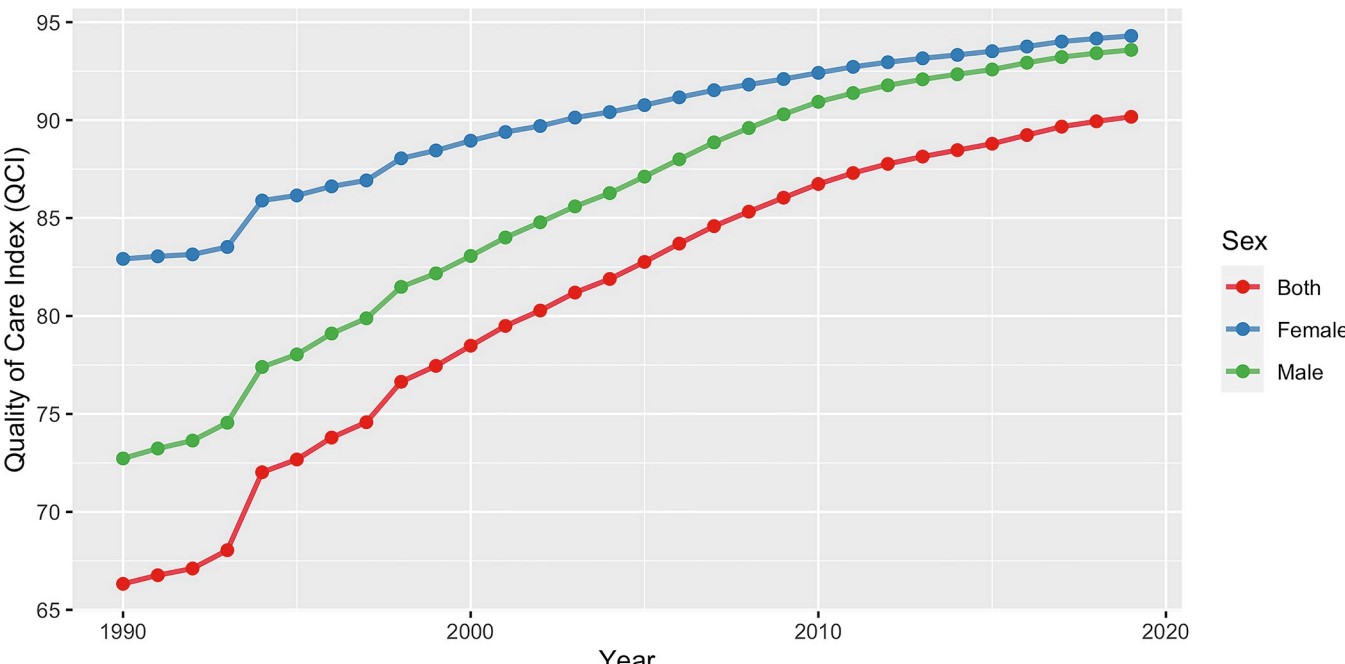

**Fig 1. The temporal trend of the age-standardized QCI for orofacial clefts, presented as a percentage, is depicted for both genders from 1990 to 2019.**
QCI: Quality of care index.

82.9 and 72.7, respectively. Over time, both males and females globally witnessed an enhancement in the quality of care, resulting in QCI scores of 93.6 for males and 94.3 for females by 2019. This indicates that the gap in care quality between the sexes narrowed down over the years (Fig 1).

When examining the GDR across various countries, it's important to note that in 1990, a significant number of African and Central Asian countries, as well as some countries in South America, exhibited a situation where males received better care than females. However, by the year 2019, a remarkable shift had occurred, and in most countries across the world, gender parity in care provision had been achieved. Refer to Fig 2 for the global distribution of GDR for both men and women in 1990 and 2017.

## Comparison between countries

Between 1990 and 2019, the QCI for orofacial clefts showed an overall increase in all regions and countries, with the exception of eight countries, where the change was less than 1%, except for Zimbabwe, which experienced a notable decrease of –4.8%; Norway had the highest QCI score of 100.0 in 1990, while Brazil had the lowest at 0.0, and in 2019, Norway maintained the highest QCI score at 99.9, while Somalia had the lowest QCI score at 59.1 (Fig 3).

In 2019, the five countries with the lowest QCI were all located in Africa, while European, North American, and Oceanian countries boasted the highest QCIs (refer to Fig 3). Globally, the countries that ranked highest for QCI in 2019 were Norway (99.9), followed by Ireland (99.4), France (99.4), Germany (99.3), and the Netherlands (99.3). In contrast, at the lower end of the QCI spectrum, we find Somalia (59.1), Niger (67.6), Burkina Faso (72.6), Ethiopia (73.0), and Mali (74.4). Fig 3 provides an extensive list of countries based on their QCIs.

When considering the World Bank Income Levels classification, high-income countries recorded the highest QCI in 2019 (98.7), while low-income countries had the lowest QCI

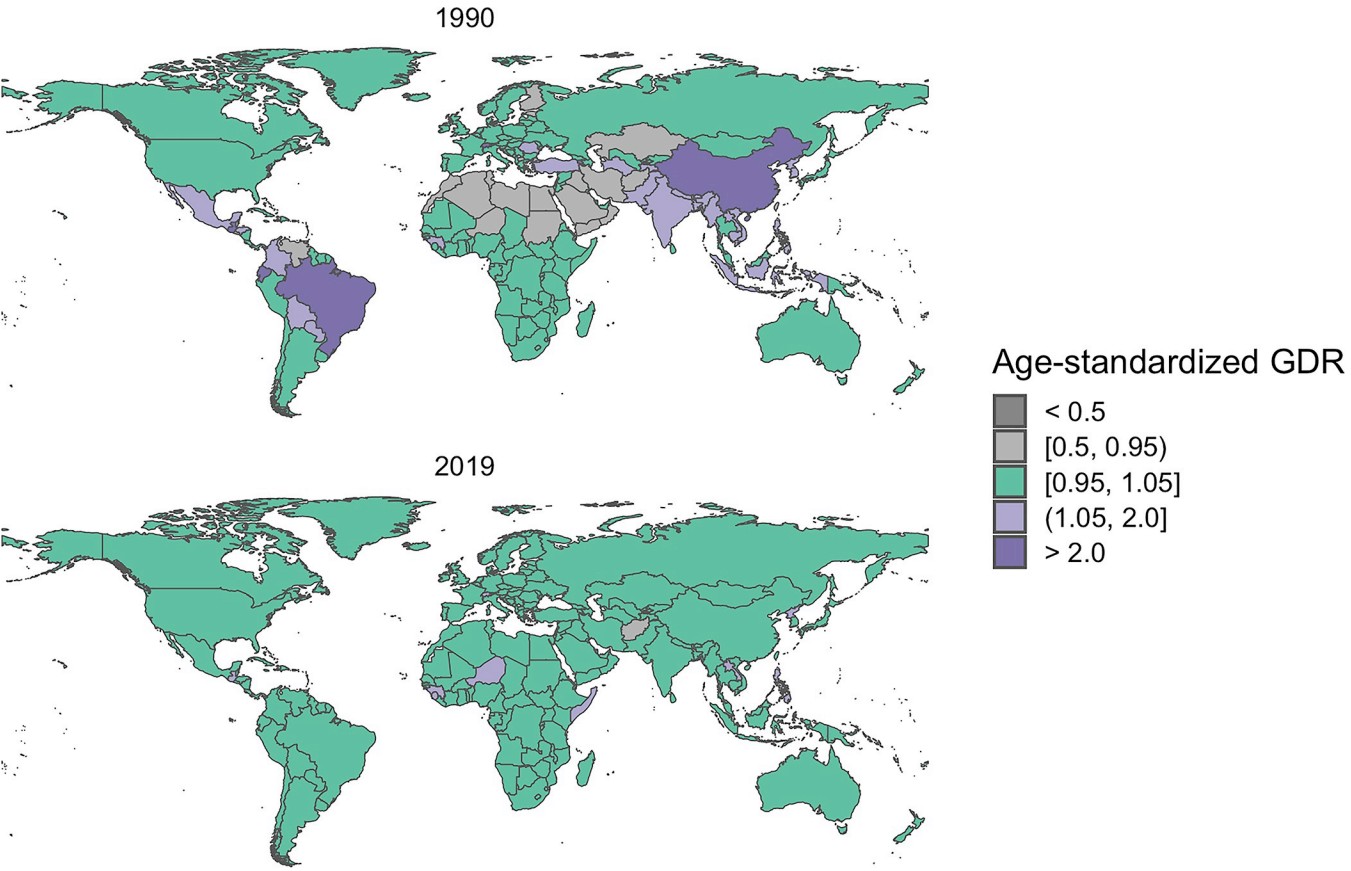

**Fig 2.** The geographical distribution of the Age-Standardized Gender Disparity Ratio (GDR) for lip and oral cavity cancer, comparing men and women in 1990 (Part A) and 2019 (Part B). GDR: Gender disparity ratio.

(80.3) for the same year. Based on the SDI classification, high SDI countries exhibited the highest QCI in 2019 (98.5), with low SDI countries having the lowest QCI (84.7) in the same year. World Bank upper-middle-income countries (54.6) and high-middle SDI countries (50.9) demonstrated the highest rate of increase in QCI (see Table 1).

In 1990, countries in Africa, Asia, and South America fell within the lowest quintile for QCI, while European and Oceanian countries occupied the highest QCI rankings. However, by 2019, African countries remained in the lowest quintile for QCI, whereas North American countries had ascended to the highest quintile, joining European and Oceanian countries (see Fig 4).

## Discussion

Our study introduces the Quality of Care Index (QCI) as a novel metric for assessing the quality of care for orofacial clefts on a global scale. By analyzing data from the Global Burden of Disease study between 1990 and 2019, we found a general improvement in the QCI worldwide, with particularly notable progress in upper-middle SDI and upper-to-middle-income regions. This trend suggests that global efforts to enhance care for individuals with orofacial clefts have yielded positive results over the past three decades, aligning with previous observations of improved outcomes in cleft care [25, 26].

The disparities in QCI scores across different regions and income levels highlight persistent inequalities in healthcare provision for orofacial clefts. High-income and high SDI countries

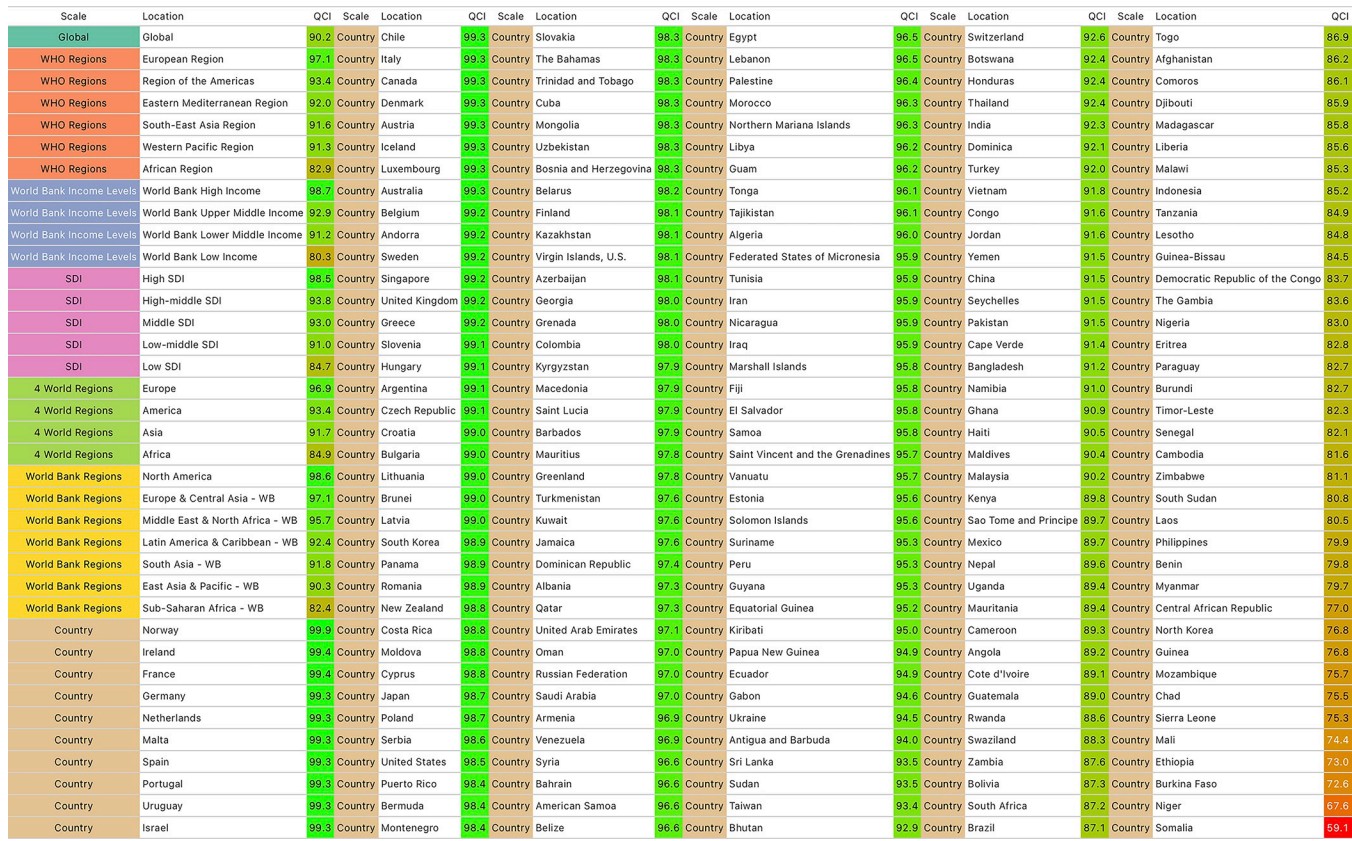

**Fig 3. Global regions and countries listed in descending order based on their QCIs in 2019.**

consistently demonstrated the highest QCI scores, while low-income and low SDI countries lagged behind. This finding underscores the critical need for targeted interventions and resource allocation to improve care quality in less developed regions, as previously emphasized by Mossey et al. [6]. The particularly low QCI score in Somalia, for instance, emphasizes the urgent need for international cooperation and support to enhance healthcare infrastructure and service delivery in such severely affected areas [27].

Our analysis revealed a converging trend in QCI between males and females globally from 1990 to 2019. This encouraging finding suggests that gender disparities in orofacial cleft care have narrowed over time, consistent with broader trends in global health equity [28]. However, the persistence of gender differences in some regions, particularly in parts of Africa, South America, and Asia, indicates that continued efforts are needed to ensure equal access to quality care regardless of gender, as highlighted by Nagem Filho et al. [29].

The development of the QCI addresses a significant gap in the literature by providing a comprehensive, quantitative measure of care quality for orofacial clefts. Unlike previous studies that often focused on specific aspects of care or were limited to particular regions [30, 31], our index offers a standardized approach to comparing care quality across countries and over time. This tool can be valuable for policymakers and healthcare planners in identifying areas for improvement and tracking progress in orofacial cleft care.

The observed decline in QCI for some high-income countries, as well as Somalia and Zimbabwe, is a concerning trend that warrants further investigation. While these countries still maintained QCI values above the global average, the downward trend suggests potential issues

**Table 1. Estimates of burden and QCI of orofacial clefts by World Bank income groups.**

| DALYs rate in 2019 (per 100,000) | DALYs rate change 1990 to 2019 | QCI in 2019 (%) | QCI change 1990 to 2019 |
|---|---|---|---|
| **World** | | | |
| 7.51 (5.10–11.57) | –12.12 | 90.20 | 23.84 |
| **World Bank Regions** | | | |
| High-income countries | | | |
| 2.04 (1.29–2.95) | –1.00 | 98.68 | 3.25 |
| Upper-middle-income countries | | | |
| 5.06 (3.88–6.57) | –25.10 | 92.88 | 54.55 |
| Lower-middle-income countries | | | |
| 9.08 (6.16–13.45) | –8.18 | 91.17 | 11.70 |
| Low-income countries | | | |
| 10.62 (5.29–22.14) | –5.12 | 80.33 | 11.26 |
| **SDI quintiles** | | | |
| High SDI quintile | | | |
| 2.20 (1.40–3.16) | –1.39 | 98.51 | 4.23 |
| High-middle SDI quintile | | | |
| 4.66 (3.50–6.10) | –22.18 | 93.85 | 50.93 |
| Middle SDI quintile | | | |
| 5.90 (4.33–7.80) | –14.55 | 93.04 | 29.67 |
| Low-middle SDI quintile | | | |
| 9.30 (6.34–13.91) | –13.00 | 90.98 | 18.19 |
| Low SDI quintile | | | |
| 10.97 (6.13–20.87) | –6.55 | 84.72 | 9.94 |

DALY: disability-adjusted life year; QCI: Quality of Care Index; SDI: Sociodemographic Index

The change was calculated as the subtraction of the value in the year 2019 from that of the year 1990.

in maintaining or improving care quality. This finding highlights the importance of continuous monitoring and improvement of healthcare systems, even in relatively well-resourced settings, as emphasized by Shkoukani et al. [32].

Our results have significant implications for global health policy. Countries with low QCI scores, particularly in Africa, should prioritize the development of comprehensive care programs for orofacial clefts. This may include investments in specialized training for healthcare providers, improving access to surgical interventions, and establishing multidisciplinary care teams, as recommended by the WHO [33]. International organizations and high-performing countries could play a crucial role in supporting these efforts through knowledge transfer, resource sharing, and capacity building initiatives [34]. The International Perinatal Database of Typical Orofacial Clefts (IPDTOC) represents a significant advancement in this regard, offering a valuable international initiative to improve the screening and management of the orofacial clefts [35]. However, it's important to note that many countries, especially in Africa and Asia, lack national registries to contribute to initiatives like IPDTOC, highlighting ongoing inequalities in care and data availability. Population-representative data are crucial for informing future policymaking regarding orofacial clefts in all countries. Additionally, addressing the availability, adequacy, and cost of necessary treatment remains a significant challenge that needs to be tackled comprehensively.

The QCI displayed a converging pattern between males and females worldwide during the 1990–2019 period, with both genders experiencing a consistent increase in QCI. The GDR

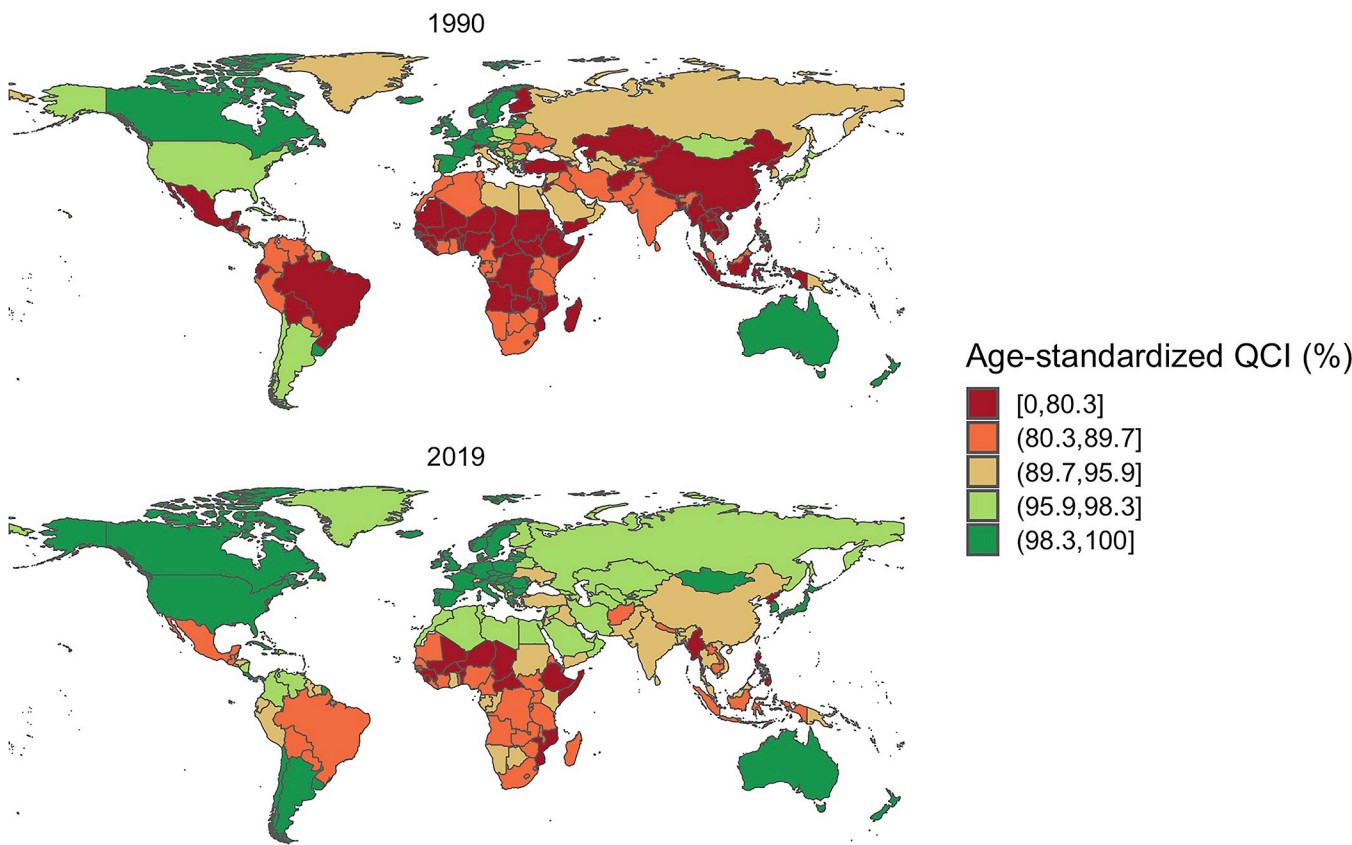

**Fig 4.** The geographic distribution of age-standardized Quality of Care Index (QCI) percentages for lip and oral cavity cancer in both men and women in 1990 (Part A) and 2019 (Part B). QCI: Quality of care index.

results indicated that African, South American, and Asian countries made progress in terms of reducing gender disparity and narrowing the gender gaps in the context of orofacial clefts.

The QCI developed in this study provides a foundation for future research in several directions. Validation studies comparing QCI scores with direct measures of care quality and patient outcomes would further strengthen the index's utility. Additionally, in-depth analyses of high-performing countries could identify best practices that could be adapted and implemented in other settings, as suggested by Chahine etl al. [36]. Future studies could also explore the relationships between QCI scores and specific healthcare system characteristics or socio-economic factors to better understand the determinants of care quality for orofacial clefts.

While our study provides valuable insights, it is important to acknowledge its limitations. The QCI is based on population-level data and may not capture individual patient experiences. Furthermore, data availability and quality vary across countries, potentially affecting the accuracy of QCI estimates in some regions, a challenge noted in previous global health studies [37]. Despite these limitations, the QCI represents a significant step forward in quantifying and comparing the quality of care for orofacial clefts globally. By highlighting disparities and tracking progress over time, this index can serve as a powerful tool for advocating for improved care and guiding efforts to reduce the global burden of orofacial clefts [38].

## Limitations

Despite its name, our index (QCI) cannot directly measure the quality of care for orofacial clefts. Given the limitations in obtaining comprehensive global, regional, and national datasets

for monitoring these malformations, we aimed to incorporate the most relevant population-level data to draw inferences and generalize results for assessing the quality of care. For instance, higher ratios of YLLs to YLDs indicate greater disease-related mortality, which may suggest a lower quality of care in a specific country or region, although it does not directly represent the quality of care. Additionally, our estimations were based on the GBD 2019 data specific to orofacial clefts. Consequently, caution should be exercised when applying our results, as they are applicable primarily at the national and global levels and not on an individual, per-patient basis. Finally, the QCI scores are relative values, limited to comparisons within the GBD 2019 database, and do not represent absolute measures of quality.

## Conclusion

The quality of care for orofacial clefts demonstrated an overall positive trend on a global scale between 1990 and 2019, with most countries experiencing an increase in QCI values for both males and females. Notably, upper-middle SDI and upper-to-middle-income regions witnessed the greatest improvements in QCI, although a few countries exhibited a declining trend during this period.

## Supporting information

**S1 Appendix. Reproducible analysis codes and their outputs.**
(PDF)

## Author Contributions

**Conceptualization:** Ahmad Sofi-Mahmudi, Erfan Shamsoddin.

**Data curation:** Ahmad Sofi-Mahmudi.

**Formal analysis:** Ahmad Sofi-Mahmudi.

**Investigation:** Ahmad Sofi-Mahmudi, Erfan Shamsoddin, Marcos Roberto Tovani-Palone.

**Methodology:** Ahmad Sofi-Mahmudi, Erfan Shamsoddin, Sahar Khademioore.

**Project administration:** Ahmad Sofi-Mahmudi.

**Resources:** Ahmad Sofi-Mahmudi.

**Software:** Ahmad Sofi-Mahmudi.

**Supervision:** Ahmad Sofi-Mahmudi.

**Validation:** Ahmad Sofi-Mahmudi, Erfan Shamsoddin, Sahar Khademioore, Marcos Roberto Tovani-Palone.

**Visualization:** Ahmad Sofi-Mahmudi, Amin Vahdati.

**Writing – original draft:** Ahmad Sofi-Mahmudi, Erfan Shamsoddin, Amin Vahdati, Marcos Roberto Tovani-Palone.

**Writing – review & editing:** Ahmad Sofi-Mahmudi, Erfan Shamsoddin, Sahar Khademioore, Yeganeh Khazaei, Amin Vahdati, Marcos Roberto Tovani-Palone.

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
