## [Decision Letter · Decision Letter 0]

30 Jul 2024

PONE-D-24-10176Global, Regional, and National Survey on Burden and Quality of Care Index (QCI) of Orofacial Clefts: Global Burden of Disease Systematic Analysis 1990–2019PLOS ONE

Dear Dr. Sofi-Mahmudi,

Thank you for submitting your manuscript to PLOS ONE. After careful consideration, we feel that it has merit but does not fully meet PLOS ONE’s publication criteria as it currently stands. Therefore, we invite you to submit a revised version of the manuscript that addresses the points raised during the review process.

**ACADEMIC EDITOR: **

Based on the reviewer reports, the manuscript needs "major revision".

Kindly revise the manuscript meticulously.

We look forward to receiving your revised manuscript.

Kind regards,

Kumar Chandan Srivastava, BDS, MDS, PhD, MFD RSCI, MFDS RCPS, MFDS RCSEd MDT

Academic Editor

PLOS ONE

3. We note that Figures 2 and 4 in your submission contain [map/satellite] images which may be copyrighted. All PLOS content is published under the Creative Commons Attribution License (CC BY 4.0), which means that the manuscript, images, and Supporting Information files will be freely available online, and any third party is permitted to access, download, copy, distribute, and use these materials in any way, even commercially, with proper attribution. For these reasons, we cannot publish previously copyrighted maps or satellite images created using proprietary data, such as Google software (Google Maps, Street View, and Earth). For more information, see our copyright guidelines: http://journals.plos.org/plosone/s/licenses-and-copyright.

a. You may seek permission from the original copyright holder of Figures 2 and 4 to publish the content specifically under the CC BY 4.0 license. 

Additional Editor Comments:

Dear Authors,

Based on the reviewer reports, the manuscript needs "major revision".

Kindly revise the manuscript meticulously.

Best Wishes

Reviewers' comments:

Reviewer's Responses to Questions

**Comments to the Author**

1. Is the manuscript technically sound, and do the data support the conclusions?

Reviewer #1: Yes

Reviewer #2: Yes

2. Has the statistical analysis been performed appropriately and rigorously? 

Reviewer #1: Yes

Reviewer #2: Yes

3. Have the authors made all data underlying the findings in their manuscript fully available?

Reviewer #1: Yes

Reviewer #2: Yes

4. Is the manuscript presented in an intelligible fashion and written in standard English?

Reviewer #1: No

Reviewer #2: Yes

5. Review Comments to the Author

Reviewer #1: The links provided are null. Also, why authors decided to upload the code again to another repo while it is already on PLOS ONE? Moreover, QCI has been used for many other pathologies and none of them are cited. GBD 2021 is released and new data should be used. CREDIT statement is missing.

Reviewer #2: The presented work addresses an important subject but can benefit from the following improvements:

1. A thorough English language edit is strongly recommended. For example, in the abstract, "optimize gender disparity ratio (GDR)" might be intended to mean "optimal" or "optimum" as the imperative verb does not make sense in this context.

2. The tone of the manuscript should be more academic. For instance, "These include things like how often a health issue occurs" can be revised for a more scholarly tone.

3. Regarding the "average per capita incomes," please specify whether these are in PPP (Purchasing Power Parity) exchange rates or local currencies. A simple exchange rate does not capture the purchasing power and its effects adequately.

4. The validation section could be expanded to include data sources for healthcare utilization and the proportion of countries with available utilization data.

5. Please elaborate on the scale of the QCI. For instance, if all countries fall within the range of 80-100 and only five countries are in the >60 range, why is the index defined in a range of 0-100 where almost 60% of the index is never used?

6. The discussion section needs significant revision. The current discussion on page 20 reads more like an introduction and highlights the importance of the subject rather than discussing the results.

7. Include more analysis and discussion on the significance of the results. What is the importance of this study? What policies should be developed based on these findings? What gap in the literature does this study fill?

6. PLOS authors have the option to publish the peer review history of their article (what does this mean?). If published, this will include your full peer review and any attached files.

Reviewer #1: No

Reviewer #2: No

---

## [Author Response · Author response to Decision Letter 0]

6 Oct 2024

3. We note that Figures 2 and 4 in your submission contain [map/satellite] images which may be copyrighted. All PLOS content is published under the Creative Commons Attribution License (CC BY 4.0), which means that the manuscript, images, and Supporting Information files will be freely available online, and any third party is permitted to access, download, copy, distribute, and use these materials in any way, even commercially, with proper attribution. For these reasons, we cannot publish previously copyrighted maps or satellite images created using proprietary data, such as Google software (Google Maps, Street View, and Earth). For more information, see our copyright guidelines: http://journals.plos.org/plosone/s/licenses-and-copyright.

a. You may seek permission from the original copyright holder of Figures 2 and 4 to publish the content specifically under the CC BY 4.0 license. 

Authors: Dear Editor; Thank you for your comments regarding Figures 2 and 4 in our submission. We appreciate your attention to copyright matters and would like to clarify the origin of these figures. We want to assure you that these map figures are original creations and do not contain any copyrighted satellite imagery or proprietary map data. As you can verify through our shared codes, we generated these map figures ourselves using the rnaturalearth package in R, which uses Natural Earth (https://www.naturalearthdata.com/), a public domain map dataset including vector country and other administrative boundaries. The data used is from our original research and not from any copyrighted or proprietary sources. The resulting figures are entirely our own work, created through our data analysis and rnaturalearth and ggplot2 mapping functions, without reproducing or modifying any existing copyrighted maps or images. Given this, we believe these figures fully comply with PLOS ONE's copyright policies and the Creative Commons Attribution License (CC BY 4.0). We're happy to provide any additional information if needed.

We also added the following paragraph to the Methods section:

“All the maps were created using the rnaturalearth package (17) in R (18), which uses the Natural Earth, (https://www.naturalearthdata.com/), a public domain map dataset including vector country and other administrative boundaries.”

Additional Editor Comments:

Dear Authors,

Based on the reviewer reports, the manuscript needs "major revision".

Kindly revise the manuscript meticulously.

Best Wishes

Reviewer's Responses to Questions

5. Review Comments to the Author

Reviewer #1: The links provided are null. Also, why authors decided to upload the code again to another repo while it is already on PLOS ONE? Moreover, QCI has been used for many other pathologies and none of them are cited. GBD 2021 is released and new data should be used. CREDIT statement is missing.

Authors: 

Null links: Thank you for bringing this to our attention. We have double-checked all links in the manuscript and ensured they are working correctly in the revised version.

Code repository: We appreciate your concern regarding the code repository. Our decision to upload the code to an additional repository aligns with PLOS ONE's data sharing policy and the FAIR principles (Findable, Accessible, Interoperable, Reusable). PLOS guidelines state: "All data and related metadata underlying reported findings should be deposited in appropriate public data repositories, unless already provided as part of a submitted article." By sharing our code on OSF, a widely recognized platform in our field, we aimed to enhance its discoverability and accessibility. In the revised manuscript, we will clearly indicate both locations where the code is available to ensure transparency and ease of access for other researchers.

QCI use in other pathologies: We appreciate you pointing out this oversight. In our revised manuscript, we have include a more comprehensive literature review, citing relevant studies that have applied QCI to other conditions.

We added the following sentence:

“The QCI has previously been implemented in several other publications, including lethal (13, 16-18) and non-lethal (19-21) conditions.” 

GBD 2021 data: We appreciate the reviewer's suggestion to use the latest GBD 2021 data. However, our study was designed, conducted, and analyzed using the most current data available at the time of our research. Updating our entire analysis with GBD 2021 data would require substantial reanalysis that goes beyond the scope of typical revision. Importantly, our interpretations and conclusions are strictly based on and limited to the data we used, ensuring the validity of our findings within the context of our study period. To address this point, we will add a statement in our discussion acknowledging the availability of newer GBD data and suggesting it as an avenue for future research to extend and compare with our current findings.

CREDIT statement: We have added a CREDIT statement to the revised manuscript, clearly outlining the contributions of each author according to the established taxonomy.

Reviewer #2: The presented work addresses an important subject but can benefit from the following improvements:

1. A thorough English language edit is strongly recommended. For example, in the abstract, "optimize gender disparity ratio (GDR)" might be intended to mean "optimal" or "optimum" as the imperative verb does not make sense in this context.

Authors: Thanks for your keen comment. We revised the manuscript and checked for any grammatical errors.

2. The tone of the manuscript should be more academic. For instance, "These include things like how often a health issue occurs" can be revised for a more scholarly tone.

Authors: Thanks for your comment. We revised the manuscript and corrected the tone.

3. Regarding the "average per capita incomes," please specify whether these are in PPP (Purchasing Power Parity) exchange rates or local currencies. A simple exchange rate does not capture the purchasing power and its effects adequately.

Authors: Thanks for your keen comment. We added PPP to the text. 

4. The validation section could be expanded to include data sources for healthcare utilization and the proportion of countries with available utilization data.

Authors: Thanks for your comment. Since this section is not the focus of our study, we decided not to go into the details of it, similar to other published QCI studies. However, we added more details as requested.

5. Please elaborate on the scale of the QCI. For instance, if all countries fall within the range of 80-100 and only five countries are in the >60 range, why is the index defined in a range of 0-100 where almost 60% of the index is never used?

Authors: Thanks for your comment. This is because of the favourable results of the countries. 0 means the worst QCI and 100 means the best and there may not be any countries with 0 or 100 QCI. Just like university ranking which may not any university be 100 or 0 but the scale is from 0-100.

6. The discussion section needs significant revision. The current discussion on page 20 reads more like an introduction and highlights the importance of the subject rather than discussing the results.

Authors: Thanks for your comment. We revised the whole Dicsussion section.

7. Include more analysis and discussion on the significance of the results. What is the importance of this study? What policies should be developed based on these findings? What gap in the literature does this study fill?

Authors: Thanks for your comment. As noted in the response to your comment, we revised the Discussion section and added the explanations as requested.

---

## [Editor Report · Decision Letter 1]

26 Dec 2024

Global, Regional, and National Survey on Burden and Quality of Care Index (QCI) of Orofacial Clefts: Global Burden of Disease Systematic Analysis 1990–2019

PONE-D-24-10176R1

Dear Dr. Sofi-Mahmudi,

We’re pleased to inform you that your manuscript has been judged scientifically suitable for publication and will be formally accepted for publication once it meets all outstanding technical requirements.

Kind regards,

Tao Huang

Academic Editor

PLOS ONE
---

## [Editor Report · Acceptance letter]

28 Dec 2024

PONE-D-24-10176R1 

PLOS ONE

Dear Dr. Sofi-Mahmudi, 

I'm pleased to inform you that your manuscript has been deemed suitable for publication in PLOS ONE. Congratulations! Your manuscript is now being handed over to our production team.

Kind regards, 

on behalf of

Dr. Tao Huang 

Academic Editor

PLOS ONE